# IMPROVING SEQUENTIAL LATENT VARIABLE MODELS WITH AUTOREGRESSIVE FLOWS

## ABSTRACT

We propose an approach for sequence modeling based on autoregressive normalizing flows. Each autoregressive transform, acting across time, serves as a moving reference frame for modeling higher-level dynamics. This technique provides a simple, general-purpose method for improving sequence modeling, with connections to existing and classical techniques. We demonstrate the proposed approach both with standalone models, as well as a part of larger sequential latent variable models. Results are presented on three benchmark video datasets, where flow-based dynamics improve log-likelihood performance over baseline models.

## 1 INTRODUCTION

Data often contain sequential structure, providing a rich signal for learning models of the world. Such models are useful for learning self-supervised representations of sequences (Li & Mandt, 2018; Ha & Schmidhuber, 2018) and planning sequences of actions (Chua et al., 2018; Hafner et al., 2019). While sequential models have a longstanding tradition in probabilistic modeling (Kalman et al., 1960), it is only recently that improved computational techniques, primarily deep networks, have facilitated learning such models from high-dimensional data (Graves, 2013), particularly video and audio. Dynamics in these models typically contain a combination of stochastic and deterministic variables (Bayer & Osendorfer, 2014; Chung et al., 2015; Gan et al., 2015; Fraccaro et al., 2016), using simple distributions (e.g. Gaussian) to directly model the likelihood of data observations. However, attempting to capture all sequential dependencies with relatively unstructured dynamics may make it more difficult to learn such models. Intuitively, the model should use its dynamical components to track *changes* in the input instead of simultaneously modeling the entire signal. Rather than expanding the computational capacity of the model, we seek a method for altering the representation of the data to provide a more structured form of dynamics.

To incorporate more structured dynamics, we propose an approach for sequence modeling based on autoregressive normalizing flows (Kingma et al., 2016; Papamakarios et al., 2017), consisting of one or more autoregressive transforms in time. A single transform is equivalent to a Gaussian autoregressive model. However, by stacking additional transforms or latent variables on top, we can arrive at more expressive models. Each autoregressive transform serves as a *moving reference frame* in which higher-level structure is modeled. This provides a general mechanism for separating different forms of dynamics, with higher-level stochastic dynamics modeled in the simplified space provided by lower-level deterministic transforms. In fact, as we discuss, this approach generalizes the technique of modeling temporal derivatives to simplify dynamics estimation (Friston, 2008).

We empirically demonstrate this approach, both with standalone autoregressive normalizing flows, as well as by incorporating these flows within more flexible sequential latent variable models. While normalizing flows have been applied in a few sequential contexts previously, we emphasize the use of these models in conjunction with sequential latent variable models. We present experimental results on three benchmark video datasets, showing improved quantitative performance in terms of log-likelihood. In formulating this general technique for improving dynamics estimation in the framework of normalizing flows, we also help to contextualize previous work.

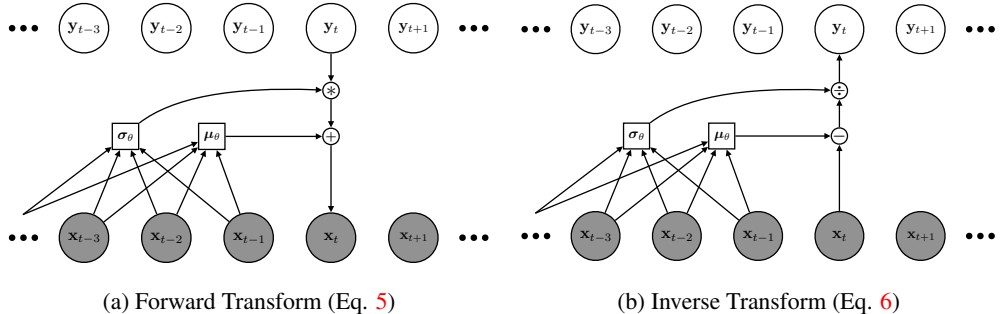

(a) Forward Transform (Eq. 5)  (b) Inverse Transform (Eq. 6)

Figure 1: **Affine Autoregressive Transforms**. Computational diagrams for forward and inverse affine autoregressive transforms (Papamakarios et al., 2017). Each $\mathbf{y}_t$ is an affine transform of $\mathbf{x}_t$, with the affine parameters potentially non-linear functions of $\mathbf{x}_{<t}$. The inverse transform is capable of converting a *correlated* input, $\mathbf{x}_{1:T}$, into a *less correlated* variable, $\mathbf{y}_{1:T}$.

## 2 BACKGROUND

### 2.1 AUTOREGRESSIVE MODELS

Consider modeling discrete sequences of observations, $\mathbf{x}_{1:T} \sim p_{\text{data}}(\mathbf{x}_{1:T})$, using a probabilistic model, $p_\theta(\mathbf{x}_{1:T})$, with parameters $\theta$. Autoregressive models (Frey et al., 1996; Bengio & Bengio, 2000) use the chain rule of probability to express the joint distribution over all time steps as the product of $T$ conditional distributions. Because of the forward nature of the world, as well as for handling variable-length sequences, these models are often formulated in forward temporal order:

$$p_\theta(\mathbf{x}_{1:T}) = \prod_{t=1}^{T} p_\theta(\mathbf{x}_t|\mathbf{x}_{<t}). \tag{1}$$

Each conditional distribution, $p_\theta(\mathbf{x}_t|\mathbf{x}_{<t})$, models the temporal dependence between time steps, i.e. a prediction of the future. For continuous variables, we often assume that each distribution takes a relatively simple form, such as a diagonal Gaussian density:

$$p_\theta(\mathbf{x}_t|\mathbf{x}_{<t}) = \mathcal{N}(\mathbf{x}_t; \boldsymbol{\mu}_\theta(\mathbf{x}_{<t}), \text{diag}(\boldsymbol{\sigma}_\theta^2(\mathbf{x}_{<t}))), \tag{2}$$

where $\boldsymbol{\mu}_\theta(\cdot)$ and $\boldsymbol{\sigma}_\theta(\cdot)$ are functions denoting the mean and standard deviation, often sharing parameters over time steps. While these functions may take the entire past sequence of observations as input, e.g. through a recurrent neural network, they may also be restricted to a convolutional window (van den Oord et al., 2016a). Autoregressive models can also be applied to non-sequential data (van den Oord et al., 2016b), where they excel at capturing local dependencies. However, due to their restrictive distributional forms, such models often struggle to capture higher-level structure.

### 2.2 AUTOREGRESSIVE LATENT VARIABLE MODELS

Autoregressive models can be improved by incorporating latent variables, often represented as a corresponding sequence, $\mathbf{z}_{1:T}$. Classical examples include Gaussian state space models and hidden Markov models (Murphy, 2012). The joint distribution, $p_\theta(\mathbf{x}_{1:T}, \mathbf{z}_{1:T})$, has the following form:

$$p_\theta(\mathbf{x}_{1:T}, \mathbf{z}_{1:T}) = \prod_{t=1}^{T} p_\theta(\mathbf{x}_t|\mathbf{x}_{<t}, \mathbf{z}_{\leq t}) p_\theta(\mathbf{z}_t|\mathbf{x}_{<t}, \mathbf{z}_{<t}). \tag{3}$$

Unlike the simple, parametric form in Eq. 2, evaluating $p_\theta(\mathbf{x}_t|\mathbf{x}_{<t})$ now requires integrating over the latent variables,

$$p_\theta(\mathbf{x}_t|\mathbf{x}_{<t}) = \int p_\theta(\mathbf{x}_t|\mathbf{x}_{<t}, \mathbf{z}_{\leq t}) p_\theta(\mathbf{z}_{\leq t}|\mathbf{x}_{<t}) d\mathbf{z}_{\leq t}, \tag{4}$$

yielding a more flexible distribution. However, performing this integration in practice is typically intractable, requiring approximate inference techniques, like variational inference (Jordan et al.,

1998). Recent works have parameterized these models with deep neural networks, e.g. (Chung et al., 2015; Gan et al., 2015; Fraccaro et al., 2016; Karl et al., 2017), using amortized variational inference (Kingma & Welling, 2014; Rezende et al., 2014) for inference and learning. Typically, the conditional likelihood, $p_\theta(\mathbf{x}_t | \mathbf{x}_{<t}, \mathbf{z}_{\leq t})$, and the prior, $p_\theta(\mathbf{z}_t | \mathbf{x}_{<t}, \mathbf{z}_{<t})$, are Gaussian densities, with temporal conditioning handled through deterministic recurrent networks and the stochastic latent variables. Such models have demonstrated success in audio (Chung et al., 2015; Fraccaro et al., 2016) and video modeling (Xue et al., 2016; Gemici et al., 2017; Denton & Fergus, 2018; He et al., 2018; Li & Mandt, 2018). However, design choices for these models remain an active area of research, with each model proposing new combinations of deterministic and stochastic dynamics.

## 2.3 Autoregressive Flows

Our approach is based on affine autoregressive normalizing flows (Kingma et al., 2016; Papamakarios et al., 2017). Here, we review this basic concept, continuing with the perspective of temporal sequences, however, it is worth noting that these flows were initially developed and demonstrated in *static* settings. Kingma et al. (2016) noted that sampling from an autoregressive Gaussian model is an invertible transform, resulting in a *normalizing flow* (Rippel & Adams, 2013; Dinh et al., 2015; 2017; Rezende & Mohamed, 2015). Flow-based models transform between simple and complex probability distributions while maintaining exact likelihood evaluation. To see their connection to autoregressive models, we can express sampling a Gaussian random variable, $\mathbf{x}_t \sim p_\theta(\mathbf{x}_t | \mathbf{x}_{<t})$ (Eq. 2), using the reparameterization trick (Kingma & Welling, 2014; Rezende et al., 2014):

$$\mathbf{x}_t = \boldsymbol{\mu}_\theta(\mathbf{x}_{<t}) + \boldsymbol{\sigma}_\theta(\mathbf{x}_{<t}) \odot \mathbf{y}_t, \tag{5}$$

where $\mathbf{y}_t \sim \mathcal{N}(\mathbf{y}_t; \mathbf{0}, \mathbf{I})$ is an auxiliary random variable and $\odot$ denotes element-wise multiplication. Thus, $\mathbf{x}_t$ is an invertible transform of $\mathbf{y}_t$, with the inverse given as

$$\mathbf{y}_t = \frac{\mathbf{x}_t - \boldsymbol{\mu}_\theta(\mathbf{x}_{<t})}{\boldsymbol{\sigma}_\theta(\mathbf{x}_{<t})}, \tag{6}$$

where division is performed element-wise. The inverse transform in Eq. 6 acts to normalize (hence, *normalizing* flow) and therefore decorrelate $\mathbf{x}_{1:T}$. Given the functional mapping between $\mathbf{y}_t$ and $\mathbf{x}_t$ in Eq. 5, the change of variables formula converts between probabilities in each space:

$$\log p_\theta(\mathbf{x}_{1:T}) = \log p_\theta(\mathbf{y}_{1:T}) - \log \left| \det \left( \frac{\partial \mathbf{x}_{1:T}}{\partial \mathbf{y}_{1:T}} \right) \right|. \tag{7}$$

By the construction of Eqs. 5 and 6, the Jacobian in Eq. 7 is triangular, enabling efficient evaluation as the product of diagonal terms:

$$\log \left| \det \left( \frac{\partial \mathbf{x}_{1:T}}{\partial \mathbf{y}_{1:T}} \right) \right| = \sum_{t=1}^{T} \sum_{i} \log \sigma_{\theta,i}(\mathbf{x}_{<t}), \tag{8}$$

where $i$ denotes the observation dimension, e.g. pixel. For a Gaussian autoregressive model, $p_\theta(\mathbf{y}_{1:T}) = \mathcal{N}(\mathbf{y}_{1:T}; \mathbf{0}, \mathbf{I})$. With these components, the change of variables formula (Eq. 7) provides an equivalent method for sampling and evaluating the model, $p_\theta(\mathbf{x}_{1:T})$, from Eqs. 1 and 2.

We can improve upon this simple set-up by chaining together multiple transforms, effectively resulting in a hierarchical autoregressive model. Letting $\mathbf{y}_{1:T}^m$ denote the variables after the $m^{\text{th}}$ transform, the change of variables formula for $M$ transforms is

$$\log p_\theta(\mathbf{x}_{1:T}) = \log p_\theta(\mathbf{y}_{1:T}^M) - \log \left| \det \left( \frac{\partial \mathbf{x}_{1:T}}{\partial \mathbf{y}_{1:T}^1} \right) \right| - \sum_{m=1}^{M-1} \log \left| \det \left( \frac{\partial \mathbf{y}_{1:T}^m}{\partial \mathbf{y}_{1:T}^{m+1}} \right) \right|. \tag{9}$$

Autoregressive flows were initially considered in the contexts of variational inference (Kingma et al., 2016) and generative modeling (Papamakarios et al., 2017). These approaches are, in fact, generalizations of previous approaches with affine transforms (Dinh et al., 2015; 2017). While autoregressive flows are well-suited for sequential data, as mentioned previously, these approaches, as well as many recent approaches (Huang et al., 2018; Oliva et al., 2018; Kingma & Dhariwal, 2018), were initially applied in static settings, such as images.

More recent works have started applying flow-based models to sequential data. For instance, van den Oord et al. (2018) and Ping et al. (2019) *distill* autoregressive speech models into flow-based models.

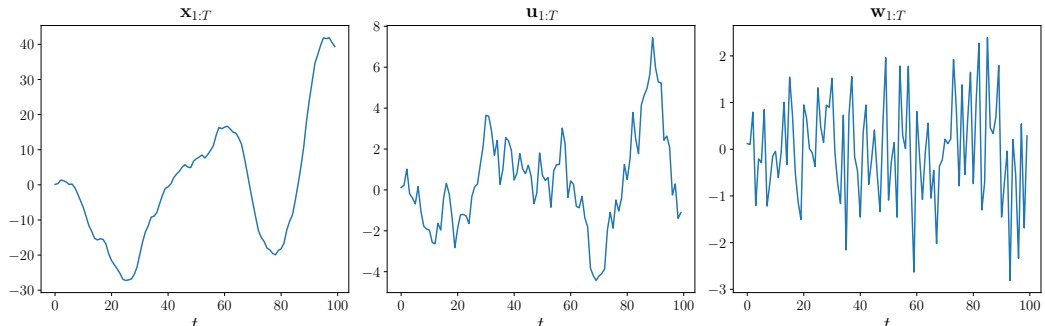

Figure 2: **Motivating Example**. Plots are shown for a sample of $\mathbf{x}_{1:T}$ (left), $\mathbf{u}_{1:T}$ (center), and $\mathbf{w}_{1:T}$ (right). Here, $\mathbf{w}_{1:T} \sim \mathcal{N}(\mathbf{w}_{1:T}; \mathbf{0}, \mathbf{I})$, and $\mathbf{u}$ and $\mathbf{x}$ are initialized at $0$. Moving from $\mathbf{x} \to \mathbf{u} \to \mathbf{w}$ via affine transforms results in successively less temporal correlation and therefore simpler dynamics.

Prenger et al. (2019) and Kim et al. (2019) instead train these models directly. Kumar et al. (2019) use a flow to model individual video frames, with an autoregressive prior modeling dynamics across time steps. Rhinehart et al. (2018) and Rhinehart et al. (2019) use autoregressive flows for modeling vehicle motion, and Henter et al. (2019) use flows for motion synthesis with motion-capture data. Ziegler & Rush (2019) learn distributions over sequences of discrete observations (e.g., text) by using flows to model dynamics of continuous latent variables. Like these recent works, we apply flow-based models to sequential data. However, we demonstrate that autoregressive flows can serve as a useful, general-purpose technique for improving sequence modeling as components of sequential latent variable models. To the best of our knowledge, our work is the first to focus on the aspect of using flows to pre-process sequential data to improve downstream dynamics modeling.

Finally, we utilize affine flows (Eq. 5) in this work. This family of flows includes methods like NICE (Dinh et al., 2015), RealNVP (Dinh et al., 2017), IAF (Kingma et al., 2016), MAF (Papamakarios et al., 2017), and GLOW (Kingma & Dhariwal, 2018). However, there has been recent work in non-affine flows (Huang et al., 2018; Jaini et al., 2019; Durkan et al., 2019), which may offer further flexibility. We chose to investigate affine flows for their relative simplicity and connections to previous techniques, however, the use of non-affine flows could result in additional improvements.

## 3 METHOD

We now describe our approach for sequence modeling with autoregressive flows. Although the core idea is a relatively straightforward extension of autoregressive flows, we show how this simple technique can be incorporated within autoregressive latent variable models (Section 2.2), providing a general-purpose approach for improving dynamics modeling. We first motivate the benefits of affine autoregressive transforms in the context of sequence modeling with a simple example.

### 3.1 A MOTIVATING EXAMPLE

Consider the discrete dynamical system defined by the following set of equations:

$$\mathbf{x}_t = \mathbf{x}_{t-1} + \mathbf{u}_t, \tag{10}$$

$$\mathbf{u}_t = \mathbf{u}_{t-1} + \mathbf{w}_t, \tag{11}$$

where $\mathbf{w}_t \sim \mathcal{N}(\mathbf{w}_t; \mathbf{0}, \boldsymbol{\Sigma})$. We can express $\mathbf{x}_t$ and $\mathbf{u}_t$ in probabilistic terms as

$$\mathbf{x}_t \sim \mathcal{N}(\mathbf{x}_t; \mathbf{x}_{t-1} + \mathbf{u}_{t-1}, \boldsymbol{\Sigma}), \tag{12}$$

$$\mathbf{u}_t \sim \mathcal{N}(\mathbf{u}_t; \mathbf{u}_{t-1}, \boldsymbol{\Sigma}). \tag{13}$$

Physically, this describes the noisy dynamics of a particle with momentum and mass 1, subject to Gaussian noise. That is, $\mathbf{x}$ represents position, $\mathbf{u}$ represents velocity, and $\mathbf{w}$ represents stochastic forces. If we consider the dynamics at the level of $\mathbf{x}$, we can use the fact that $\mathbf{u}_{t-1} = \mathbf{x}_{t-1} - \mathbf{x}_{t-2}$ to write

$$p(\mathbf{x}_t|\mathbf{x}_{t-1}, \mathbf{x}_{t-2}) = \mathcal{N}(\mathbf{x}_t; \mathbf{x}_{t-1} + \mathbf{x}_{t-1} - \mathbf{x}_{t-2}, \boldsymbol{\Sigma}). \tag{14}$$

Thus, we see that in the space of $\mathbf{x}$, the dynamics are second-order Markov, requiring knowledge of the past two time steps. However, at the level of $\mathbf{u}$ (Eq. 13), the dynamics are first-order Markov, requiring only the previous time step. Yet, note that $\mathbf{u}_t$ is, in fact, an affine autoregressive transform of $\mathbf{x}_t$ because $\mathbf{u}_t = \mathbf{x}_t - \mathbf{x}_{t-1}$ is a special case of the general form $\frac{\mathbf{x}_t - \boldsymbol{\mu}_\theta(\mathbf{x}_{<t})}{\boldsymbol{\sigma}_\theta(\mathbf{x}_{<t})}$. In Eq. 10, we see that the Jacobian of this transform is $\partial \mathbf{x}_t / \partial \mathbf{u}_t = \mathbf{I}$, so, from the change of variables formula, we have $p(\mathbf{x}_t | \mathbf{x}_{t-1}, \mathbf{x}_{t-2}) = p(\mathbf{u}_t | \mathbf{u}_{t-1})$. In other words, an affine autoregressive transform has allowed us to convert a second-order Markov system into a first-order Markov system, thereby simplifying the dynamics. Continuing this process to move to $\mathbf{w}_t = \mathbf{u}_t - \mathbf{u}_{t-1}$, we arrive at a representation that is entirely temporally decorrelated, i.e. no dynamics, because $p(\mathbf{w}_t) = \mathcal{N}(\mathbf{w}_t; \mathbf{0}, \boldsymbol{\Sigma})$. A sample from this system is shown in Figure 2, illustrating this process of temporal decorrelation.

The special case of modeling temporal changes, $\mathbf{u}_t = \mathbf{x}_t - \mathbf{x}_{t-1} = \Delta \mathbf{x}_t$, is a common pre-processing technique; for recent examples, see Deisenroth et al. (2013); Chua et al. (2018); Kumar et al. (2019). In fact, $\Delta \mathbf{x}_t$ is a finite differences approximation of the generalized velocity (Friston, 2008) of $\mathbf{x}$, a classic modeling technique in dynamical models and control (Kalman et al., 1960), redefining the state-space to be first-order Markov. Affine autoregressive flows offer a generalization of this technique, allowing for non-linear transform parameters and flows consisting of multiple transforms, with each transform serving to successively decorrelate the input sequence in time. In analogy with generalized velocity, each transform serves as a *moving reference frame*, allowing us to focus model capacity on less correlated fluctuations rather than the highly temporally correlated raw signal.

## 3.2 AUTOREGRESSIVE FLOWS ON SEQUENCES

We apply autoregressive flows across time steps within a sequence, $\mathbf{x}_{1:T} \in \mathbb{R}^{T \times D}$. That is, the observation at each time step, $\mathbf{x}_t \in \mathbb{R}^D$, is modeled as an autoregressive function of past observations, $\mathbf{x}_{<t} \in \mathbb{R}^{t-1 \times D}$, and a random variable, $\mathbf{y}_t \in \mathbb{R}^D$ (Figure 3a). We consider flows of the form given in Eq. 5, where $\boldsymbol{\mu}_\theta(\mathbf{x}_{<t})$ and $\boldsymbol{\sigma}_\theta(\mathbf{x}_{<t})$ are parameterized by neural networks. In constructing chains of flows, we denote the shift and scale functions at the $m^{\text{th}}$ transform as $\boldsymbol{\mu}_\theta^m(\cdot)$ and $\boldsymbol{\sigma}_\theta^m(\cdot)$ respectively. We then calculate $\mathbf{y}^m$ using the corresponding inverse transform:

$$\mathbf{y}_t^m = \frac{\mathbf{y}_t^{m-1} - \boldsymbol{\mu}_\theta^m(\mathbf{y}_{<t}^{m-1})}{\boldsymbol{\sigma}_\theta^m(\mathbf{y}_{<t}^{m-1})}. \tag{15}$$

After the final ($M^{\text{th}}$) transform, the base distribution, $p_\theta(\mathbf{y}_{1:T}^M)$, can range from a simple distribution, e.g. $\mathcal{N}(\mathbf{y}_{1:T}^M; \mathbf{0}, \mathbf{I})$, in the case of a flow-based model, up to more complicated distributions in the case of other latent variable models (Section 3.3). While flows of greater depth can improve model capacity, such transforms have limiting drawbacks. In particular, 1) they require that the outputs maintain the same dimensionality as the inputs, $\mathbb{R}^{T \times D}$, 2) they are restricted to affine transforms, and 3) these transforms operate element-wise within a time step. As we discuss in the next section, we can combine autoregressive flows with non-invertible sequential latent variable models (Section 2.2), which do not have these restrictions.

## 3.3 LATENT VARIABLE MODELS WITH AUTOREGRESSIVE FLOWS

We can use autoregressive flows as a component in parameterizing the dynamics within autoregressive latent variable models. To simplify notation, we consider this set-up with a single transform, but a chain of multiple transforms (Section 3.2) can be applied within each flow.

### 3.3.1 MODEL FORMULATION

Let us consider parameterizing the conditional likelihood, $p_\theta(\mathbf{x}_t | \mathbf{x}_{<t}, \mathbf{z}_{\leq t})$, within a latent variable model using an autoregressive flow (Figure 3b). To do so, we express a base conditional distribution for $\mathbf{y}_t$, denoted as $p_\theta(\mathbf{y}_t | \mathbf{y}_{<t}, \mathbf{z}_{\leq t})$, which is then transformed into $\mathbf{x}_t$ via the affine transform in Eq. 5. We have written $p_\theta(\mathbf{y}_t | \mathbf{y}_{<t}, \mathbf{z}_{\leq t})$ with conditioning on $\mathbf{y}_{<t}$, however, by removing temporal correlations to arrive at $\mathbf{y}_{1:T}$, our hope is that these dynamics can be primarily modeled through $\mathbf{z}_{1:T}$. Using the change of variables formula, we can express the latent variable model's log-joint distribution as

$$\log p_\theta(\mathbf{x}_{1:T}, \mathbf{z}_{1:T}) = \log p_\theta(\mathbf{y}_{1:T}, \mathbf{z}_{1:T}) - \log \left| \det \left( \frac{\partial \mathbf{x}_{1:T}}{\partial \mathbf{y}_{1:T}} \right) \right|, \tag{16}$$

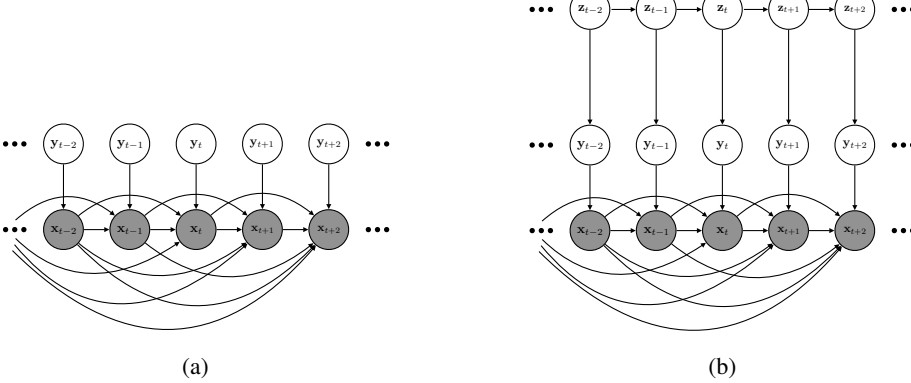

(a)                                            (b)

Figure 3: **Graphical Models**. Diagrams for **(a)** a single-transform affine autoregressive flow-based model, with random variables, $\mathbf{y}_{1:T} \sim \mathcal{N}(\mathbf{y}_{1:T}; \mathbf{0}, \mathbf{I})$, and **(b)** a sequential latent variable model with a flow-based conditional likelihood. We have depicted simplified dynamics for $\mathbf{y}_{1:T}$ and $\mathbf{z}_{1:T}$ for clarity, however, in general, these can be non-Markov. The flow removes low-level temporal correlations in $\mathbf{x}_{1:T}$, whereas the latent variables, $\mathbf{z}_{1:T}$, capture any remaining structure in $\mathbf{y}_{1:T}$.

where the joint distribution over $\mathbf{y}_{1:T}$ and $\mathbf{z}_{1:T}$, in general, is given as

$$p_\theta(\mathbf{y}_{1:T}, \mathbf{z}_{1:T}) = \prod_{t=1}^{T} p_\theta(\mathbf{y}_t|\mathbf{y}_{<t}, \mathbf{z}_{\leq t}) p_\theta(\mathbf{z}_t|\mathbf{y}_{<t}, \mathbf{z}_{<t}). \tag{17}$$

Note that the latent prior, $p_\theta(\mathbf{z}_t|\mathbf{y}_{<t}, \mathbf{z}_{<t})$, can be equivalently conditioned on $\mathbf{x}_{<t}$ or $\mathbf{y}_{<t}$, as there is a one-to-one mapping between these variables. We could also consider parameterizing the prior with autoregressive flows, or even constructing a hierarchy of latent variables. However, we leave these extensions for future work, opting to first introduce the basic concept here.

### 3.3.2 VARIATIONAL INFERENCE & LEARNING

Training a latent variable model via maximum likelihood requires marginalizing over the latent variables to evaluate the marginal log-likelihood of observations: $\log p_\theta(\mathbf{x}_{1:T}) = \log \int p_\theta(\mathbf{x}_{1:T}, \mathbf{z}_{1:T}) d\mathbf{z}_{1:T}$. This marginalization is typically intractable, requiring the use of approximate inference methods. Variational inference (Jordan et al., 1998) introduces an approximate posterior distribution, $q(\mathbf{z}_{1:T}|\mathbf{x}_{1:T})$, which provides a lower bound on the marginal log-likelihood:

$$\log p_\theta(\mathbf{x}_{1:T}) \geq \mathcal{L}(\mathbf{x}_{1:T}; q, \theta) \equiv \mathbb{E}_{q(\mathbf{z}_{1:T}|\mathbf{x}_{1:T})} \left[ \log p_\theta(\mathbf{x}_{1:T}, \mathbf{z}_{1:T}) - \log q(\mathbf{z}_{1:T}|\mathbf{x}_{1:T}) \right], \tag{18}$$

referred to as the evidence lower bound (ELBO). Often, we assume $q(\mathbf{z}_{1:T}|\mathbf{x}_{1:T})$ is a *structured* distribution, attempting to explicitly capture the model's temporal dependencies across $\mathbf{z}_{1:T}$. We can consider both *filtering* or *smoothing* inference, however, we focus on the case of filtering, with

$$q(\mathbf{z}_{1:T}|\mathbf{x}_{1:T}) = \prod_{t=1}^{T} q(\mathbf{z}_t|\mathbf{x}_{\leq t}, \mathbf{z}_{<t}). \tag{19}$$

The conditional dependencies in $q$ can be modeled through a direct, amortized function, e.g. using a recurrent network (Chung et al., 2015), or through optimization (Marino et al., 2018). Again, note that we can condition $q$ on $\mathbf{x}_{\leq t}$ or $\mathbf{y}_{\leq t}$, as there exists a one-to-one mapping between these variables. With the model's joint distribution (Eq. 16) and approximate posterior (Eq. 19), we can then evaluate the ELBO. We derive the ELBO for this set-up in Appendix A, yielding

$$\mathcal{L} = \sum_{t=1}^{T} \mathbb{E}_{q(\mathbf{z}_{\leq t}|\mathbf{y}_{\leq t})} \left[ \log p_\theta(\mathbf{y}_t|\mathbf{y}_{<t}, \mathbf{z}_{\leq t}) - \log \frac{q(\mathbf{z}_t|\mathbf{y}_{\leq t}, \mathbf{z}_{<t})}{p_\theta(\mathbf{z}_t|\mathbf{y}_{<t}, \mathbf{z}_{<t})} - \log \left| \det \left( \frac{\partial \mathbf{x}_t}{\partial \mathbf{y}_t} \right) \right| \right]. \tag{20}$$

This expression makes it clear that a flow-based conditional likelihood amounts to learning a latent variable model on top of the intermediate learned space provided by $\mathbf{y}$, with an additional factor in the objective penalizing the scaling between $\mathbf{x}$ and $\mathbf{y}$.

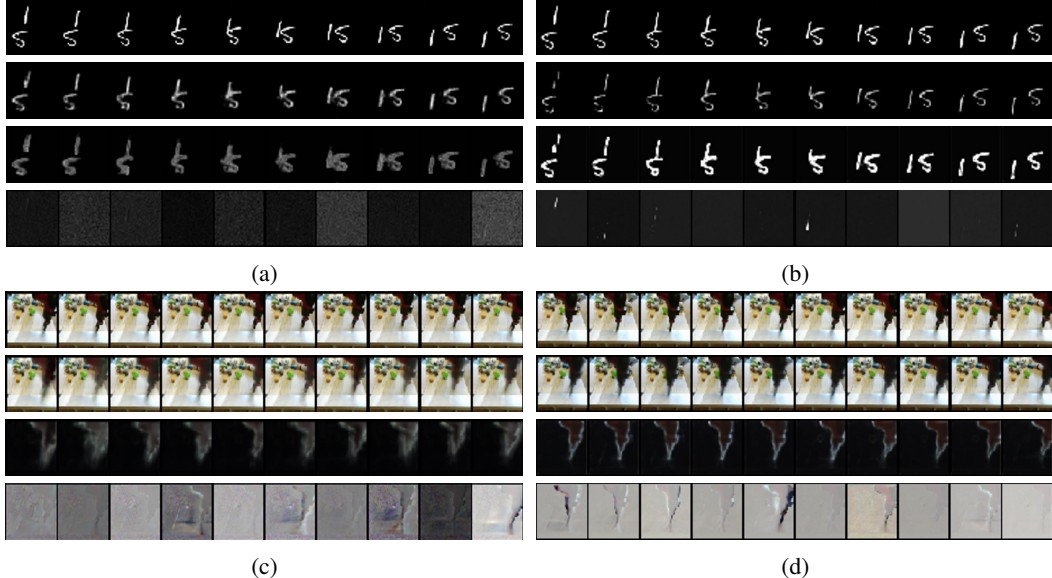

Figure 4: **Flow Visualization**. Visualization of the flow component for **(a), (c)** standalone flow-based models and **(b), (d)** sequential latent variable models with flow-based conditional likelihoods for Moving MNIST and BAIR Robot Pushing. From top to bottom, each figure shows 1) the original frames, $\mathbf{x}_t$, 2) the predicted shift, $\boldsymbol{\mu}_\theta(\mathbf{x}_{<t})$, for the frame, 3) the predicted scale, $\boldsymbol{\sigma}_\theta(\mathbf{x}_{<t})$, for the frame, and 4) the noise, $\mathbf{y}_t$, obtained from the inverse transform.

## 4 EVALUATION

We demonstrate and evaluate the proposed framework on three benchmark video datasets: Moving MNIST (Srivastava et al., 2015), KTH Actions (Schuldt et al., 2004), and BAIR Robot Pushing (Ebert et al., 2017). Experimental setups are described in Section 4.1, followed by a set of qualitative experiments in Section 4.2. In Section 4.3, we provide quantitative comparisons across different model classes. Further implementation details and visualizations can be found in Appendix B. Anonymized code is available at the following link.

### 4.1 EXPERIMENTAL SETUP

We implement three classes of models: 1) standalone autoregressive flow-based models, 2) sequential latent variable models, and 3) sequential latent variable models with flow-based conditional likelihoods. Flows are implemented with convolutional networks, taking in a fixed window of previous frames and outputting shift, $\boldsymbol{\mu}_\theta$, and scale, $\boldsymbol{\sigma}_\theta$, parameters. The sequential latent variable models consist of convolutional and recurrent networks for both the *encoder* and *decoder* networks, following the basic form of architecture that has been previously employed in video modeling (Denton & Fergus, 2018; Ha & Schmidhuber, 2018; Hafner et al., 2019).

In the case of a regular sequential latent variable model, the conditional likelihood is a Gaussian that models the frame, $\mathbf{x}_t$. In the case of a flow-based conditional likelihood, we model the noise variable, $\mathbf{y}_t$, with a Gaussian. In our experiments, the flow components have vastly fewer parameters than the sequential latent variable models. In addition, for models with flow-based conditional likelihoods, we restrict the number of parameters to enable a fairer comparison. These models have *fewer* parameters than the baseline sequential latent variable models (with non-flow-based conditional likelihoods). See Appendix B for parameter comparisons and architecture details. Finally, flow-based conditional likelihoods only add a constant computational cost per time-step, requiring a single forward pass per time step for both evaluation and generation.

Table 1: **Quantitative Comparison.** Average test log-likelihood (higher is better) in *nats per pixel per channel* for Moving MNIST, BAIR Robot Pushing, and KTH Actions. For flow-based models (1-AF and 2-AF), we report the average log-likelihood. For sequential latent variable models (SLVM and SLVM w/ 1-AF), we report the average lower bound on the log-likelihood.

|  | M-MNIST | BAIR | KTH |
|---|---|---|---|
| 1-AF | $-2.15$ | $-3.05$ | $-3.34$ |
| 2-AF | $-2.13$ | $-2.90$ | $-3.35$ |
| SLVM | $\geq -1.92$ | $\geq -3.57$ | $\geq -4.63$ |
| SLVM w/ 1-AF | $\geq \mathbf{-1.86}$ | $\geq \mathbf{-2.35}$ | $\geq \mathbf{-2.39}$ |

## 4.2 QUALITATIVE EVALUATION

To better understand the behavior of autoregressive flows on sequences, we visualize each component as an image. In Figure 4, we show the data, $\mathbf{x}_t$, shift, $\boldsymbol{\mu}_\theta$, scale, $\boldsymbol{\sigma}_\theta$, and noise variable, $\mathbf{y}_t$, for standalone flow-based models (left) and flow-based conditional likelihoods (right) on random sequences from the Moving MNIST and BAIR Robot Pushing datasets. Similar visualizations for KTH Actions are shown in Figure 8 in the Appendix. In Figure 9 in the Appendix, we also visualize these quantities for a flow-based conditional likelihood with two transforms.

From these visualizations, we can make a few observations. The shift parameters (second row) tend to capture the static background, blurring around regions of uncertainty. The scale parameters (third row), on the other hand, tend to focus on regions of higher uncertainty, as expected. The resulting noise variables (bottom row) display any remaining structure not modeled by the flow. In comparing standalone flow-based models with flow-based conditional likelihoods in sequential latent variable models, we see that the latter qualitatively contains more structure in $\mathbf{y}$, e.g. dots (Figure 4b, fourth row) or sharper edges (Figure 4d, fourth row). This is expected, as the noise distribution is more expressive in this case. With a relatively simple dataset, like Moving MNIST, a single flow can reasonably decorrelate the input, yielding white noise images (Figure 4a, fourth row). However, with natural image datasets like KTH Actions and BAIR Robot Pushing, a large degree of structure is still present in these images, motivating the use of additional model capacity to model this signal. In Appendix C.1, we quantify the degree of temporal decorrelation performed by flow-based models by evaluating the empirical correlation between frames at successive time steps for both the data, $\mathbf{x}$, and the noise variables, $\mathbf{y}$. In Appendix C.2, we provide additional qualitative results.

## 4.3 QUANTITATIVE EVALUATION

Log-likelihood results for each model class are shown in Table 1. We report the average test log-likelihood in *nats per pixel per channel* for flow-based models and the lower bound on this quantity for sequential latent variable models. Standalone flow-based models perform surprisingly well, even outperforming sequential latent variable models in some cases. Increasing flow depth from 1 to 2 generally results in improved performance. Sequential latent variable models with flow-based conditional likelihoods outperform their baseline counterparts, despite having *fewer* parameters. One reason for this disparity is overfitting. Comparing with the training performance reported in Table 3, we see that sequential latent variable models with flow-based conditional likelihoods overfit less. This is particularly apparent on KTH Actions, which contains training and test sets with a high degree of separation (different identities and activities). This suggests that removing static components, like backgrounds, yields a reconstruction space that is better for generalization.

The quantitative results in Table 1 are for a representative sequential latent variable model with a standard convolutional encoder-decoder architecture and fully-connected latent variables. However, many previous works do not evaluate proper lower bounds on log-likelihood, using techniques like down-weighting KL divergences (Denton & Fergus, 2018; Ha & Schmidhuber, 2018; Lee et al., 2018). Indeed, Marino et al. (2018) train SVG (Denton & Fergus, 2018) with a proper lower bound and report a lower bound of $-2.86$ *nats per pixel* on KTH Actions, on-par with our results. Kumar et al. (2019) report log-likelihood results on BAIR Robot Pushing, obtaining $-1.3$ *nats per pixel*, substantially higher than our results. However, their model is significantly larger than the models presented here, consisting of 3 levels of latent variables, each containing 24 steps of flows.

## 5 CONCLUSION

We have presented a technique for improving sequence modeling based on autoregressive normalizing flows. This technique uses affine transforms to temporally decorrelate sequential data, thereby simplifying the estimation of dynamics. We have drawn connections to classical approaches, which involve modeling temporal derivatives. Finally, we have empirically shown how this technique can improve sequential latent variable models.

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

# A  LOWER BOUND DERIVATION

Consider the model defined in Section 3.3.1, with the conditional likelihood parameterized with autoregressive flows. That is, we parameterize

$$\mathbf{x}_t = \boldsymbol{\mu}_\theta(\mathbf{x}_{<t}) + \boldsymbol{\sigma}_\theta(\mathbf{x}_{<t}) \odot \mathbf{y}_t \tag{21}$$

yielding

$$p_\theta(\mathbf{x}_t|\mathbf{x}_{<t}, \mathbf{z}_{\leq t}) = p_\theta(\mathbf{y}_t|\mathbf{y}_{<t}, \mathbf{z}_{\leq t}) \left| \det\left(\frac{\partial \mathbf{x}_t}{\partial \mathbf{y}_t}\right) \right|^{-1}. \tag{22}$$

The joint distribution over all time steps is then given as

$$p_\theta(\mathbf{x}_{1:T}, \mathbf{z}_{1:T}) = \prod_{t=1}^{T} p_\theta(\mathbf{x}_t|\mathbf{x}_{<t}, \mathbf{z}_{\leq t}) p_\theta(\mathbf{z}_t|\mathbf{x}_{<t}, \mathbf{z}_{<t}) \tag{23}$$

$$= \prod_{t=1}^{T} p_\theta(\mathbf{y}_t|\mathbf{y}_{<t}, \mathbf{z}_{\leq t}) \left| \det\left(\frac{\partial \mathbf{x}_t}{\partial \mathbf{y}_t}\right) \right|^{-1} p_\theta(\mathbf{z}_t|\mathbf{x}_{<t}, \mathbf{z}_{<t}). \tag{24}$$

To perform variational inference, we consider a filtering approximate posterior of the form

$$q(\mathbf{z}_{1:T}|\mathbf{x}_{1:T}) = \prod_{t=1}^{T} q(\mathbf{z}_t|\mathbf{x}_{\leq t}, \mathbf{z}_{<t}). \tag{25}$$

We can then plug these expressions into the evidence lower bound:

$$\mathcal{L} \equiv \mathbb{E}_{q(\mathbf{z}_{1:T}|\mathbf{x}_{1:T})} \left[ \log p_\theta(\mathbf{x}_{1:T}, \mathbf{z}_{1:T}) - \log q(\mathbf{z}_{1:T}|\mathbf{x}_{1:T}) \right] \tag{26}$$

$$= \mathbb{E}_{q(\mathbf{z}_{1:T}|\mathbf{x}_{1:T})} \left[ \log \left( \prod_{t=1}^{T} p_\theta(\mathbf{y}_t|\mathbf{y}_{<t}, \mathbf{z}_{\leq t}) \left| \det\left(\frac{\partial \mathbf{x}_t}{\partial \mathbf{y}_t}\right) \right|^{-1} p_\theta(\mathbf{z}_t|\mathbf{x}_{<t}, \mathbf{z}_{<t}) \right) \right.$$

$$\left. - \log \left( \prod_{t=1}^{T} q(\mathbf{z}_t|\mathbf{x}_{\leq t}, \mathbf{z}_{<t}) \right) \right] \tag{27}$$

$$= \mathbb{E}_{q(\mathbf{z}_{1:T}|\mathbf{x}_{1:T})} \left[ \sum_{t=1}^{T} \log p_\theta(\mathbf{y}_t|\mathbf{y}_{<t}, \mathbf{z}_{\leq t}) - \log \frac{q(\mathbf{z}_t|\mathbf{x}_{\leq t}, \mathbf{z}_{<t})}{p_\theta(\mathbf{z}_t|\mathbf{x}_{<t}, \mathbf{z}_{<t})} - \log \left| \det\left(\frac{\partial \mathbf{x}_t}{\partial \mathbf{y}_t}\right) \right| \right]. \tag{28}$$

Finally, in the filtering setting, we can rewrite the expectation, bringing it inside of the sum (see Gemici et al. (2017); Marino et al. (2018)):

$$\mathcal{L} = \sum_{t=1}^{T} \mathbb{E}_{q(\mathbf{z}_{\leq t}|\mathbf{x}_{\leq t})} \left[ \log p_\theta(\mathbf{y}_t|\mathbf{y}_{<t}, \mathbf{z}_{\leq t}) - \log \frac{q(\mathbf{z}_t|\mathbf{x}_{\leq t}, \mathbf{z}_{<t})}{p_\theta(\mathbf{z}_t|\mathbf{x}_{<t}, \mathbf{z}_{<t})} - \log \left| \det\left(\frac{\partial \mathbf{x}_t}{\partial \mathbf{y}_t}\right) \right| \right]. \tag{29}$$

Because there exists a one-to-one mapping between $\mathbf{x}_{1:T}$ and $\mathbf{y}_{1:T}$, we can equivalently condition the approximate posterior and the prior on $\mathbf{y}$, i.e.

$$\mathcal{L} = \sum_{t=1}^{T} \mathbb{E}_{q(\mathbf{z}_{\leq t}|\mathbf{y}_{\leq t})} \left[ \log p_\theta(\mathbf{y}_t|\mathbf{y}_{<t}, \mathbf{z}_{\leq t}) - \log \frac{q(\mathbf{z}_t|\mathbf{y}_{\leq t}, \mathbf{z}_{<t})}{p_\theta(\mathbf{z}_t|\mathbf{y}_{<t}, \mathbf{z}_{<t})} - \log \left| \det\left(\frac{\partial \mathbf{x}_t}{\partial \mathbf{y}_t}\right) \right| \right]. \tag{30}$$

## B EXPERIMENT DETAILS

We store a fixed number of past frames in the buffer of each transform, to generate the shift and scale for the transform. For each stack of flow, 4 convolutional layers with kernel size $(3, 3)$, stride 1 and padding 1 are applied first on each data observation in the buffer, preserving the data shape. The outputs are concatenated along the channel dimension and go through another four convolutional layers also with kernel size $(3, 3)$, stride 1 and padding 1. Finally, separate convolutional layers with the same kernel size, stride and padding are used to generate shift and scale respectively.

For latent variable models, we use a DC-GAN structure (Radford et al., 2015), with 4 layers of convolutional layers of kernel size $(4, 4)$, stride 2 and padding 1 before another convolutional layer of kernel size $(4, 4)$, stride 1 and no padding to encode the data. The encoded data is sent to an LSTM (Hochreiter & Schmidhuber, 1997) followed by fully connected layers to generate the mean and log-variance for estimating the approximate posterior distribution of the latent variable, $\mathbf{z}_t$. The conditional prior distribution is modeled with another LSTM followed by fully connected layers, taking the previous latent variable as input. The decoder take the inverse structure of the encoder. In the SLVM, we use 2 LSTM layers for modelling the conditional prior and approximate posterior distributions, while in the combined model we use 1 LSTM layer for each.

We use the Adam optimizer (Kingma & Ba, 2014) with a learning rate of $1 \times 10^{-4}$ to train all the models. For Moving MNIST, we use a batch size of 16 and train for $200,000$ iterations for latent variable models and $100,000$ iterations for flow-based and latent variable models with flow-based likelihoods. For BAIR Robot Pushing, we use a batch size of 8 and train for $200,000$ iterations for all models. For KTH dataset we use a batch size of 8 and train for $90,000$ iterations for all models. Batch norm (Ioffe & Szegedy, 2015) is applied to all convolutional layers that do not directly generate distribution or transform parameters. We randomly crop sequence of length 13 from all sequences and evaluate on the last 10 frames. (For 2-flow models we crop sequence of length 16 to fill up all buffers.) Anonymized code is available at the following link.

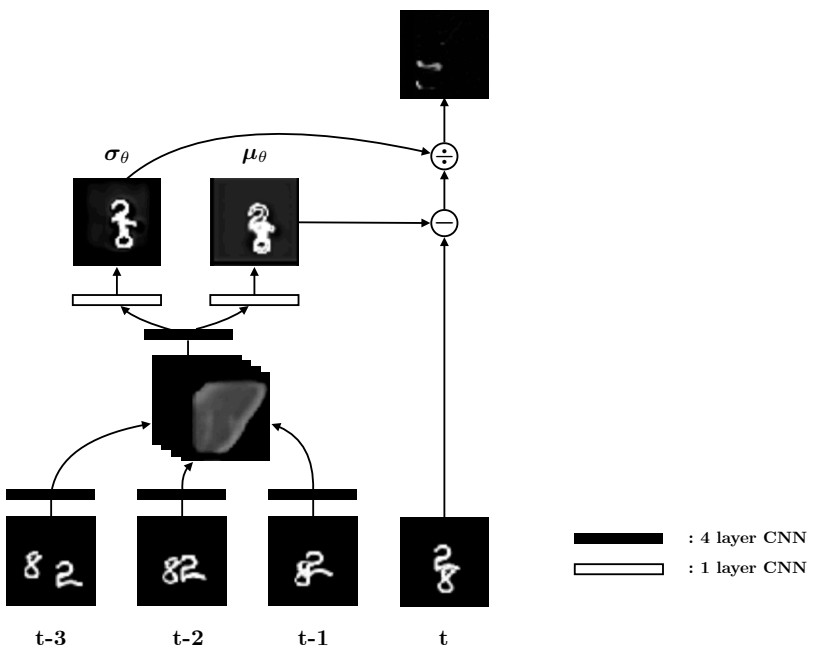

All convolution layers are set up with with kernel size (3,3) stride 1 and padding 1.

Figure 5: Implementation Visualization of the autoregressive flow.

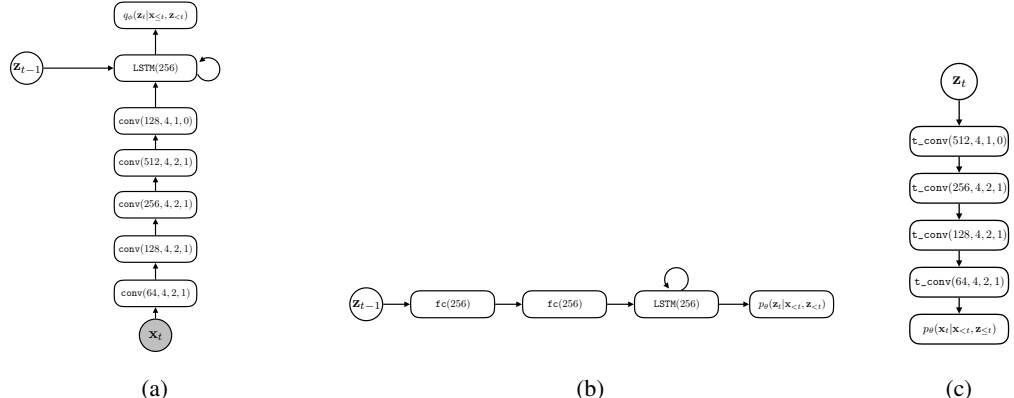

(a)    (b)    (c)

Figure 6: **Model Architecture Diagrams**. Diagrams are shown for the (**a**) approximate posterior, (**b**) conditional prior, and (**c**) conditional likelihood of the sequential latent variable model. `conv` denotes a convolutional layer, `LSTM` denotes a long short-term memory layer, `fc` denotes a fully-connected layer, and `t_conv` denotes a transposed convolutional layer. For `conv` and `t_conv` layers, the numbers in parentheses respectively denote the number of filters, filter size, stride, and padding of the layer. For `fc` and `LSTM` layers, the number in parentheses denotes the number of units.

Table 2: **Number of parameters for each model on each dataset.** Flow-based models contain relatively few parameters as compared with the SLVM, as our flows consist primarily of $3 \times 3$ convolutions with limited channels. In the SLVM, we use 2 LSTM layers for modelling the prior and posterior distribution of latent variable while in the combined model we use 1 LSTM layer for each.

| Model | 1-AF | 2-AF | SLVM | SLVM w/ 1-AF |
|---|---|---|---|---|
| Moving Mnist | 343k | 686k | 11302k | 10592k |
| BAIR Robot Pushing | 363k | 726k | 11325k | 10643k |
| KTH Action | 343k | 686k | 11302k | 10592k |

## C    ADDITIONAL EXPERIMENTAL RESULTS

### C.1    QUANTITATIVE EVALUATION OF TEMPORAL DECORRELATION

The qualitative results in Figures 4 and 8 demonstrate that flows are capable of removing much of the structure of the observations, resulting in *whitened* noise images. To quantitatively confirm the temporal decorrelation resulting from this process, we evaluate the empirical correlation between successive frames, averaged over spatial locations and channels, for the data observations and noise variables. This is an average normalized version of the *auto-covariance* of each signal with a time delay of 1 time step. Specifically, we estimate the temporal correlation as

$$\text{corr}_{\mathbf{x}} \equiv \frac{1}{C * W * H} \cdot \sum_{i,j,k}^{H,W,C} \mathbb{E}_{x_t^{(i,j,k)}, x_{t+1}^{(i,j,k)} \sim \mathcal{D}} \left[ \frac{(x_t^{(i,j,k)} - \mu^{(i,j,k)})(x_{t+1}^{(i,j,k)} - \mu^{(i,j,k)})}{(\sigma^{(i,j,k)})^2} \right], \quad (31)$$

where $x^{(i,j,k)}$ denotes the value of the image at location $(i,j)$ and channel $k$, $\mu^{(i,j,k)}$ denotes the mean of this dimension, and $\sigma^{(i,j,k)}$ denotes the standard deviation of this dimension. $H, W,$ and $C$ respectively denote the height, width, and number of channels of the observations.

We evaluated this quantity for data examples, $\mathbf{x}$, and noise variables, $\mathbf{y}$, for SLVM w/ 1-AF. The results for training sequences are shown in Table 4. In Figure 7, we plot this quantity during training for KTH Actions. We see that flows do indeed result in a decrease in temporal correlation. Note that because correlation is a measure of *linear* dependence, one cannot conclude from these results alone

Table 3: **Training Quantitative Comparison.** Average **training** log-likelihood (higher is better) in *nats per pixel per channel* for Moving MNIST, BAIR Robot Pushing, and KTH Actions. For flow-based models (1-AF and 2-AF), we report the average log-likelihood. For sequential latent variable models (SLVM and SLVM w/ 1-AF), we report the average lower bound on the log-likelihood.

|  | M-MNIST | BAIR | KTH |
|---|---|---|---|
| 1-AF | $-2.06$ | $-2.98$ | $-2.95$ |
| 2-AF | $-2.04$ | $-2.76$ | $-2.95$ |
| SLVM | $\geq -1.93$ | $\geq -3.46$ | $\geq -3.05$ |
| SLVM w/ 1-AF | $\geq \mathbf{-1.85}$ | $\geq \mathbf{-2.31}$ | $\geq \mathbf{-2.21}$ |

Table 4: **Temporal Correlation.** Temporal correlation (Eq. 31) between successive time steps for data observations, $\mathbf{x}$, and noise variables, $\mathbf{y}$, for SLVM w/ 1-AF.

|  | M-MNIST | BAIR | KTH |
|---|---|---|---|
| $\text{corr}_{\mathbf{x}}$ | 0.24 | 0.87 | 0.96 |
| $\text{corr}_{\mathbf{y}}$ | 0.02 | 0.43 | 0.31 |

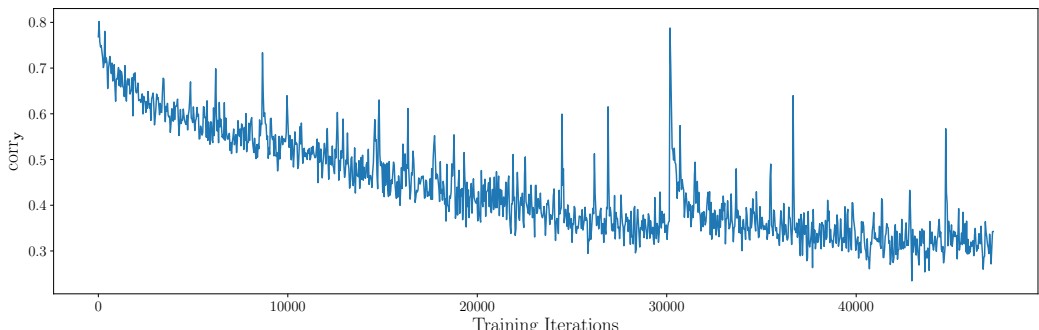

Figure 7: **Temporal Correlation During Training**. $\text{corr}_{\mathbf{y}}$ during training for SLVM w/ 1-AF on the KTH Actions. Temporal correlation decreases substantially during training.

that the flows have resulted in simplified temporal structure. However, these results agree with the qualitative and quantitative results presented in Section 4, suggesting that autoregressive flows can yield sequences with simpler dynamics.

## C.2 ADDITIONAL QUALITATIVE RESULTS

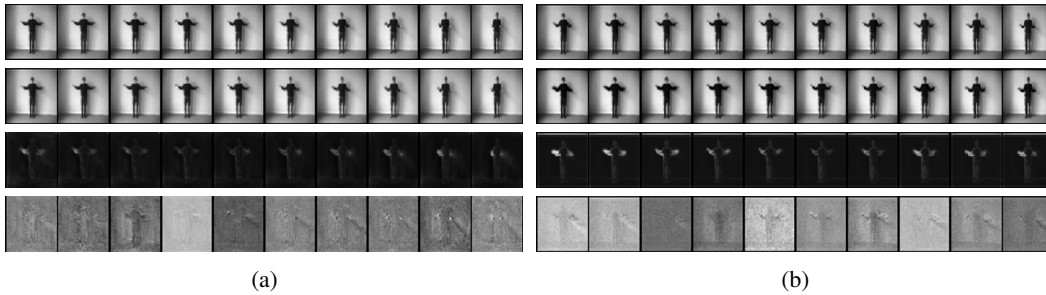

(a)                                                                                         (b)

Figure 8: **Flow Visualization on KTH Action**. Visualization of the flow component for **(a)** standalone flow-based models and **(b)** sequential latent variable models with flow-based conditional likelihoods for KTH Actions. From top to bottom, each figure shows 1) the original frames, $\mathbf{x}_t$, 2) the predicted shift, $\boldsymbol{\mu}_\theta(\mathbf{x}_{<t})$, for the frame, 3) the predicted scale, $\boldsymbol{\sigma}_\theta(\mathbf{x}_{<t})$, for the frame, and 4) the noise, $\mathbf{y}_t$, obtained from the inverse transform.

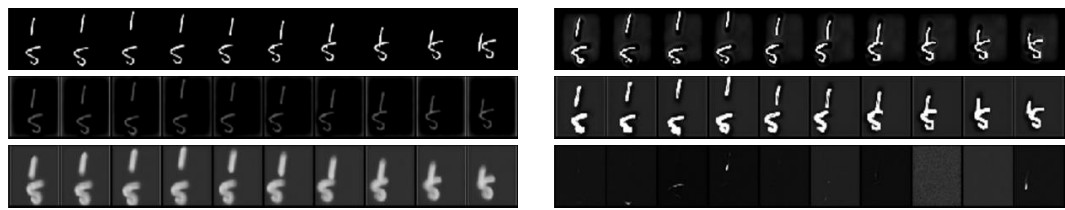

Figure 9: **SLVM w/ 2-AF Visualization on Moving MNIST**. Visualization of the flow component for sequential latent variable models with 2-layer flow-based conditional likelihoods for Moving MNIST. From top to bottom on the left side, each figure shows 1) the original frames, $\mathbf{x}_t$, 2) the lower-level predicted shift, $\boldsymbol{\mu}_\theta^1(\mathbf{x}_{<t})$, for the frame, 3) the predicted scale, $\boldsymbol{\sigma}_\theta^1(\mathbf{x}_{<t})$, for the frame. On the right side, from top to bottom, we have 1) the higer-level predicted shift, $\boldsymbol{\mu}_\theta^2(\mathbf{x}_{<t})$, for the frame, 3) the predicted scale, $\boldsymbol{\sigma}_\theta^2(\mathbf{x}_{<t})$, for the frame and 4) the noise, $\mathbf{y}_t$, obtained from the inverse transform.

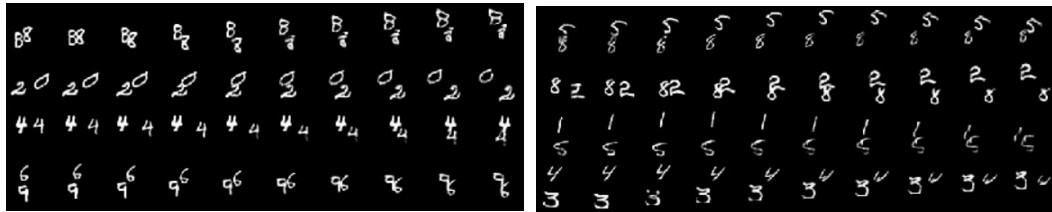

Figure 10: **Generated Moving MNIST Samples**. Samples frame sequences generated from a 2-AF model.

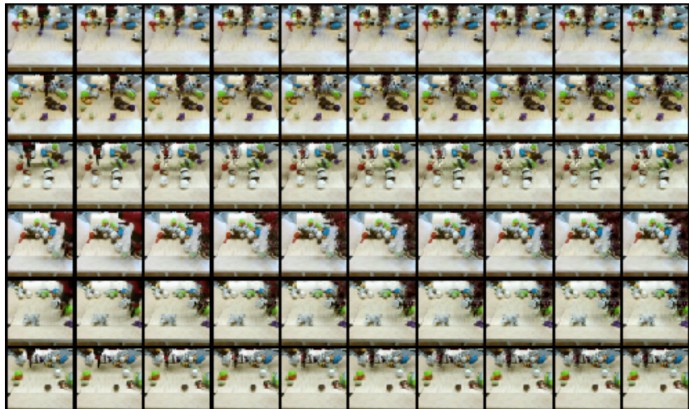

Figure 11: **Generated BAIR Robot Pushing Samples**. Samples frame sequences generated from SLVM w/ 1-AF.

