# OpenReview forum: "Improving Sequential Latent Variable Models with Autoregressive Flows"
_ICLR.cc/2020/Conference — Reject_

### Official Review · AnonReviewer3 · 2019-10-16
**Official Blind Review #3**

**Rating:** 3

**Review:**

Summary:

The paper discusses ways to use autoregressive flows in sequence modelling. Two main variants are considered:
(a) An affine autoregressive flow directly modelling the data.
(b) An affine autoregressive flow whose base distribution is a sequential VAE; equivalently, a sequential VAE whose decoder is an affine autoregressive flow.

Pros:

The paper is very well written and crystal clear. I particularly appreciated the motivating example that shows how each layer of an affine autoregressive flow reduces the order of a linear dynamical system by 1, and the connections with modelling temporal changes and moving reference frames.

The methods are technically correct and well-motivated. The experiments are done well.

Overall, the paper scores high on writing and technical quality.

Cons:

In my opinion, the paper scores low on novelty and original contribution.

In general, it's not clear to me what the claimed contribution is. More specifically:

Is the claimed contribution new methodology for modelling sequences? In my opinion, using flows as VAE decoders, or adding latent variables to a flow model and training it variationally, are standard applications of existing techniques and I wouldn't consider them particularly novel.

Is the claimed contribution improved modelling performance? The main results are that (a) replacing Gaussian decoders with autoregressive flows improves performance, and (b) adding latent variables to the base distribution of an affine autoregressive flow also improves performance. Both of these results are exactly what one would expect from our experience with these methods. Other than that, the paper doesn't present any results that indicate the particular models used enable us to do things we couldn't do before, or improve against the state of the art in sequence modelling.

Is the claimed contribution useful representations? The motivation for using the flow in this particular way as a VAE decoder is that the flow will model low-level correlations whereas the latent variables will capture high-level dynamics. However, the experiments (e.g. the visualizations) don't support this claim, and the usefulness of the learned representations hasn't been demonstrated in an alternative way,

Decision:

Even though the paper is technically correct and well written, my decision is weak reject because of the lack of novelty and original contribution.

Suggestions for improvement:

My main suggestion to the authors is to keep up the good work, but also reflect on what the specific contribution of the paper is, and try to make a stronger case for it. Some minor suggestions/corrections follow:

Eq. (8): As written, the expression makes little sense as \sigma is a vector. I understand that there is supposed to be a sum over the elements of log\sigma, so I'd suggest expressing that more clearly.

Eq. (9): It seems to me that the last Jacobian is upside down.

In general, it would be good to be more thorough on how this paper is similar to related work and how it differs. There is also this related work which may be good to discuss:

Latent Normalizing Flows for Discrete Sequences, https://arxiv.org/abs/1901.10548

In the particle analogy of the motivating example of section 3.1, it would be good to say explicitly that x is the position, u is the velocity and w is the force, to make the example even more intuitive.

The paper only considers affine autoregressive flows, but there has been a lot of recent work on non-affine autoregressive flows that are more expressive, for example:

Neural Autoregressive Flows, https://arxiv.org/abs/1804.00779
Sum-Of-Squares Polynomial Flow, https://arxiv.org/abs/1905.02325
Neural Spline Flows, https://arxiv.org/abs/1906.04032

Such flows could improve the experimental results of the paper. At the very least, it would be good to discuss them as more flexible alternatives.

In section 3.2, a third and very significant limitation of the flows discussed here is that they act elementwise on the dimensions (e.g. pixels) of y_t.

In the experimental section, it would be good to describe on a high level what the architecture of the VAE is, especially the architecture of the prior and the encoder, and the types of distributions used there (e.g. diagonal Gaussians or otherwise).

It would be good to show samples from the models in the experimental results.

**Experience Assessment:**

I have published in this field for several years.

**Review Assessment: Checking Correctness Of Derivations And Theory:**

I carefully checked the derivations and theory.

**Review Assessment: Checking Correctness Of Experiments:**

I carefully checked the experiments.

**Review Assessment: Thoroughness In Paper Reading:**

I read the paper thoroughly.

---

> ### Author Response · Authors · 2019-11-15
> **Response to Reviewer 3**
>
> Thank you for your comments! Here, we will attempt to address additional specific points:
>
> “Is the claimed contribution new methodology for modeling sequences? In my opinion, using flows as VAE decoders, or adding latent variables to a flow model and training it variationally, are standard applications of existing techniques and I wouldn't consider them particularly novel.”
>
> As mentioned, flows and VAE models have been combined in various ways (though, in our opinion, this is still under-explored), and we do not claim to introduce this combination. While affine autoregressive flows are a popular class of flow-based models, to the best of our knowledge, the application of these flows across time steps for the purposes of simplifying video modeling is novel. Specifically, our main contribution is identifying flows as a useful technique for pre-processing sequences to simplify downstream modeling.
>
> “Is the claimed contribution improved modeling performance? The main results are that (a) replacing Gaussian decoders with autoregressive flows improves performance, and (b) adding latent variables to the base distribution of an affine autoregressive flow also improves performance. Both of these results are exactly what one would expect from our experience with these methods.”
>
> Improved modeling performance is one of the results of our method. It does seem that including more flexible model components should obviously improve performance. However, from the perspective from a sequential latent variable model, it is unclear where to incorporate dynamics. For example, one could include more latent variables or recurrent networks at various stages. We specifically propose using autoregressive flows as a type of ‘pre-processing’ stage, resulting in a new sequence with dynamics that are hopefully easier to model. In our experiments, we attempt to control for the complexity of each model.
>
> “Is the claimed contribution useful representations?”
>
> This is not something that we investigated in this paper, although we intend to investigate this more thoroughly.
>
> “Eq. (8): As written, the expression makes little sense as \sigma is a vector. I understand that there is supposed to be a sum over the elements of log\sigma, so I'd suggest expressing that more clearly.”
>
> You are correct. This has been updated in the submission. Thank you!
>
> “Eq. (9): It seems to me that the last Jacobian is upside down.”
>
> Indeed. Thanks again!
>
> “In the particle analogy of the motivating example of section 3.1, it would be good to say explicitly that x is the position, u is the velocity and w is the force, to make the example even more intuitive.”
>
> We have stated this in the updated submission.
>
> “The paper only considers affine autoregressive flows, but there has been a lot of recent work on non-affine autoregressive flows that are more expressive…At the very least, it would be good to discuss them as more flexible alternatives.”
>
> We have included a discussion of these non-affine flows in the updated submission. We chose affine flows for their relative simplicity while still yielding reasonable performance.
>
> “In section 3.2, a third and very significant limitation of the flows discussed here is that they act elementwise on the dimensions (e.g. pixels) of y_t.”
>
> We have stated this limitation more specifically. However, for the purposes of removing correlations across time, rather than space, they are useful.
>
> “In the experimental section, it would be good to describe on a high level what the architecture of the VAE is, especially the architecture of the prior and the encoder, and the types of distributions used there (e.g. diagonal Gaussians or otherwise).”
>
> We have included a more thorough discussion of the model architectures, as well as diagrams in Appendix B.

---

### Official Review · AnonReviewer2 · 2019-10-17
**Official Blind Review #2**

**Rating:** 6

**Review:**


Summary
The paper proposes to combine the video modeling approaches based on autoregressive flows (e.g. Kumar’19) with amortized variational inference (e.g. Denton’18), wherein an autoregressive latent variable model optimized with variational inference is extended with an autoregressive flow that further transforms the output of the latent variable model while allowing to compute exact conditional probability. This is motivated with a physical intuition, where a dynamics model can benefit from decorrelating the inputs, and it is demonstrated that layers of autoregressive flows can represent derivatives of the original signal. In a proof-of-concept experiment, it is shown that using a layer of autoregressive flow improves NLL of a latent variable model.

Decision
The paper presents an interesting method and tackles an important problem. At the same time, the properties of the proposed method are not well exposed and the experimental evaluation is incomplete. Moreover, the motivation of the paper is confusingly disconnected from the proposed model. I rate this paper as borderline, but am hopeful that some of the issues will be clarified during the discussion period.

Pros
- The paper is well-motivated and tackles a significant problem.
- The proposed method is novel.
- The paper is well-written.

Cons
- The experimental evaluation is incomplete and does not expose the properties of the method fully. Comparisons to prior art are missing. (see below)
- The motivation is disconnected from the proposed model. The introduction of the paper motivates a model that hierarchically decorrelates a sequence of frames to arrive at a fully factorized model, which is later motivated with a physical example. However, the method proposed in the paper is instead a single layer of autoregressive flow on top of a powerful latent variable model! This is expressed in the title, but only glossed over in the abstract and introduction. The writing has to be updated to coherently focus on the contribution of the paper.

Questions (ordered by decreasing importance)
1. In table 1, quantitative results are reported for the introduced methods. It is shown that introducing autoregressive flows achieves better likelihood and better generalization. However, quantitative comparisons with published methods that were evaluated on these datasets are missing, such as Denton’18 and Kumar’19. A quick calculation shows that Kumar et al. achieves a log-likelihood of -0.43 in Table 1 when converted to this paper’s metric, although it is possible my conversion is incorrect. Is the presented model competitive with previously published results?
2. No qualitative generation results are presented. Since the model achieves a high likelihood it is likely to do well on one-frame prediction, and possibly would even work on autoregressive multi-step prediction. Is the model capable of generation of diverse and plausible video?
3. The paper has a lengthy section 3.1 that convincingly explains that decorrelating latent variables in time is important for sequence modeling. However the proposed approach in fact produces latents that are correlated in time! Since the prior over latent variables is conditioned on past frames, the model can in fact learn a correlated representation and still achieve optimal likelihood. Moreover, the position of both the digit and the robot arm could be seen in what should be the decorrelated image in Fig 4. Is there solid quantitative (or even qualitative) evidence that the model learns a ‘more decorrelated’ representation beyond the fact that it copies the background and that the likelihood improves? The evaluation in this paper does not convince me that the model learns a temporally decorrelated representation.
4. Were modern techniques beyond affine flows considered, such as from Kingma’18, Kumar’19? Two layers of affine flows are likely insufficient to model the complexity of these data, which makes the comparison to the purely flow-based models somewhat unfair.
5. It is stated that the paper is “the first to demonstrate flows across time steps for video data”, however, the related work by Kumar et al. proposes a somewhat similar model in which conditional flows are used to model video data. Do Kumar et al. not “demonstrate flows across time steps”?

Minor comments
1. Eq (10) and (12) seem to be inconsistent. Perhaps x_t = x_t-1 + u_t-1 was meant in eq (10)?
2. Line before eq(14): it not true that u_t-1 = x_t-1 - x_t-2. It would be true if the deterministic x_t = x_t-1 + u_t-1 model was assumed instead of the gaussian N(x_t; x_t-1 + u_t-1, Sigma). It is possible that eq(14) is still correct as the variance of Gaussians is additive.
3. The following work uses autoregressive flows for modeling temporal dynamics and should be cited: Rhinehart’18,19

Rhinehart et al, Deep Imitative Models for Flexible Inference, Planning, and Control
Rhinehart et al, PRECOG: PREdiction Conditioned On Goals in Visual Multi-Agent Settings

--------------------- Update 11.19 -----------------------
The newly provided experiments support some of the claims of the paper. In particular, I appreciate the plot showing that the proposed method successfully learns a more decorrelated representation over time, and the provided qualitative samples from the model. The authors also clarified my questions about motivation. At the same time, the proposed method is not shown to compare well to state-of-the-art approaches. I am leaning towards accepting the paper, but I believe the method would have a much larger impact if its properties were more fully exposed.

== comparison with Denton&Fergus'18 (SVG) ==
When trained with beta=1, as the authors suggest for comparison, this method is known to perform poorly. There are two possible ways of alleviating this: 1) to train with the modified objective as in the paper but evaluate the true lower bound on the likelihood, or 2) interpret the beta as the fixed variance of the decoder distribution. Given the results the authors have provided, I believe the latter option will lead to SVG outperforming the proposed approach.

== Correlation plot ==
Thanks for performing this experiment! While measuring correlation only captures linear dependencies, which is likely mostly the background image, this plot shows that the model indeed learns to (linearly) decorrelate the frames in the sequence.

== Samples ==
Thanks for providing samples from the model! While the performance on BAIR is not quite convincing, the MNIST samples look very good.

= Kumar et al. comparison ==
The author's response convinces me that the proposed model is significantly different from Kumar et al. in scope, as Kumar et al simply use a per-frame normalizing flow encoder coupled with a sequential prior.

== eqs. 10, 12 ==
The authors' response cleared my confusion, the equations are correct.

**Experience Assessment:**

I have published one or two papers in this area.

**Review Assessment: Checking Correctness Of Derivations And Theory:**

I carefully checked the derivations and theory.

**Review Assessment: Checking Correctness Of Experiments:**

I carefully checked the experiments.

**Review Assessment: Thoroughness In Paper Reading:**

I read the paper thoroughly.

---

> ### Author Response · Authors · 2019-11-15
> **Response to Reviewer 2**
>
> Thank you for your comments! Here, we will attempt to address additional specific points:
>
> “The paper has a lengthy section 3.1 that convincingly explains that decorrelating latent variables in time is important for sequence modeling. However the proposed approach in fact produces latents that are correlated in time!… Is there solid quantitative (or even qualitative) evidence that the model learns a ‘more decorrelated’ representation”
>
> It should be noted that while these flows have the capability of removing temporal correlations, they may not be able to remove all temporal dependencies. Thus, it can still be beneficial to model these dependencies in the base distribution of the flow. The motivation is not that we want to remove all temporal dependencies, but rather that we would like to remove as much as possible to simplify modeling for the sequential latent variable model. Based on your suggestion, we have provided a quantitative confirmation that the result of the flow is less temporally correlated than the input.
>
> “Were modern techniques beyond affine flows considered, such as from Kingma’18, Kumar’19? Two layers of affine flows are likely insufficient to model the complexity of these data, which makes the comparison to the purely flow-based models somewhat unfair.”
>
> It is important to note that we are applying flows across time steps, rather than within a time step. If we were to apply a method like GLOW (which is affine) in an analogous way, this would involve applying the flow on half the time steps as a function of the other half of the steps. Methods like VideoFlow apply flows within time steps. While this may further improve performance by removing spatial correlations, this is not the motivation of our work. Many flows are part of the general family of affine flows (e.g. NICE, RealNVP, IAF, MAF, GLOW), so we felt this was an important place to start in developing this technique. We included comparisons with standalone flow-based models to demonstrate that these models work well as generative models on their own. Note that a single affine autoregressive flow is exactly equivalent to an autoregressive model, which can perform quite well in practice.
>
> “Do Kumar et al. not “demonstrate flows across time steps”?”
>
> While Kumar et al. do apply flows within a sequential context, we mean to distinguish between applying flows within a time step vs. across time steps, as in our work. Kumar et al. use non-flow-based models to model temporal dependencies. Other works, such as van den Oord et al. with audio data, do use flows across time steps, as we do here.
>
> “Eq (10) and (12) seem to be inconsistent. Perhaps x_t = x_t-1 + u_t-1 was meant in eq (10)?”
>
> We understand the point of confusion, however, Eqs. 10 and 12 are consistent. In Eq. 10, x_t = x_t-1 + u_t gives the exact value of x_t, but in Eq. 12, x_t-1 + u_t-1 gives the mean of the Gaussian distribution over x_t. This can be seen by plugging the random variable for u_t, i.e. Eq. 13, into Eq. 10.
>
> “Line before eq(14): it not true that u_t-1 = x_t-1 - x_t-2. It would be true if the deterministic x_t = x_t-1 + u_t-1 model was assumed instead of the gaussian N(x_t; x_t-1 + u_t-1, Sigma). It is possible that eq(14) is still correct as the variance of Gaussians is additive.”
>
> This follows directly from the definition of u. To be clear, x and u are simply different ways of expressing the same randomness, subject to different offset values. This is because the transform between u_t and x_t is deterministic. All of the stochasticity originates from w_t.
>
> “The following work uses autoregressive flows for modeling temporal dynamics and should be cited: Rhinehart’18,19”
>
> Thank you for these references. We have included them in the updated draft.

---

### Official Review · AnonReviewer1 · 2019-10-22
**Official Blind Review #1**

**Rating:** 6

**Review:**

This paper proposes to model temporal sequences using autoregressive flows across time steps, that allow to model more explicitly temporal changes of the input, i.e. how the input x_t has changed w.r.t x_{<t}. As also stated by the authors, this is a generalization of other work that instead of modelling the input at each time step, models temporal differences between consecutive time steps.
To the best of my knowledge, this is the first work that models normalizing flows in the sequential setting in this way (to be fair however, the idea is fairly obvious).

Overall I found the paper interesting, and I think it is well written, so I am leaning towards acceptance. My biggest concern in the paper is the experimental section that could be improved in several ways:
- the paper misses broader perfoemance comparisons against other state of the art models, in particular videoflow which is quite related to the models introduced in this paper.
- how does the model perform on longer sequences, e.g. for long term generation? I would expect that such a direct dependence of the temporal dynamics on the frames of the video may make it hard for the model to coherently predict future latent states for many time steps.
- What would happen if we used the same trick of modelling the conditional likelihood in this way in other SOTA models?
- what are the computational requirements of the models presented in this paper?

**Experience Assessment:**

I have published in this field for several years.

**Review Assessment: Checking Correctness Of Derivations And Theory:**

I assessed the sensibility of the derivations and theory.

**Review Assessment: Checking Correctness Of Experiments:**

I assessed the sensibility of the experiments.

**Review Assessment: Thoroughness In Paper Reading:**

I read the paper at least twice and used my best judgement in assessing the paper.

---

> ### Author Response · Authors · 2019-11-15
> **Response to Reviewer 1**
>
> Thank you for your comments! Here, we will attempt to address additional specific points:
>
> “the paper misses broader performance comparisons against other state of the art models, in particular videoflow which is quite related to the models introduced in this paper.”
>
> As discussed in our common response, our goal was not to propose a specific video-modeling architecture, but rather to propose a technique for improving sequence modeling. VideoFlow applies flows within each time step, unlike our proposed technique, which operates across time steps. VideoFlow is also significantly larger than the models that we investigated, consisting of 3 levels of latent variables, each with 24 steps of flow, and each flow containing 5 residual blocks. In contrast, the models in our experiments consist of just one or two flows, with each component of our models parameterized with relatively simple convolutional or recurrent networks. As stated in our common response, our quantitative results are on-par with log-likelihood estimates for previous works, like SVG (Denton & Fergus, 2018).
>
> “What would happen if we used the same trick of modeling the conditional likelihood in this way in other SOTA models?”
>
> We chose a representative sequential latent variable model for our experiments. However, we suspect this technique will apply broadly to many sequence modeling settings. Indeed, as we noted in our submission, VideoFlow models differences in variables, which we discuss as a special case of our technique. Before the camera-ready deadline, we intend to conduct additional experiments applying our technique to some of these previously proposed models.
>
> “what are the computational requirements of the models presented in this paper?”
>
> Autoregressive flows, in the sequential context, add only a constant computational cost to each time step, requiring only a single forward pass for evaluation and generation.

---

### Author Response · Authors · 2019-11-15
**Response to All Reviewers**

We would like to thank the reviewers for their feedback; we found their comments insightful and constructive. We were also happy to see that the reviewers found the idea ‘crystal clear.’ The draft has been updated to reflect their comments. We will attempt to address common points in this post, with separate comments to each reviewer addressing specific points.

—Comparison with prior work

The motivation behind our work is to provide a general-purpose technique for improving sequence modeling. To that end, we are not proposing a specific model architecture, instead focusing on relative improvements over a representative video modeling architecture. The sequential latent variable model architecture that we used for conducting experiments is a fairly standard convolutional encoder-decoder architecture with fully-connected latent variables. This architecture resembles previous works like world models (Ha & Schmidhuber, 2018) and SVG (Denton & Fergus, 2018).

However, the difficulty in comparing video modeling performance is that many previous works employ a variety of custom techniques. For instance, many previous works do not train or evaluate their models with proper log-likelihood (or lower bound) objectives, e.g. Ha & Schmidhuber, 2018 and Denton & Fergus, 2018 both down-weight the KL term in the objective, yielding an improper lower bound. These previous works also evaluate squared pixel error, implicitly setting the std. dev. to 1. Indeed, when SVG is trained with a variational bound, the results are comparable with our reported results, e.g. -2.86 nats/dim (SVG) vs. -2.39 nats/dim (ours) on KTH Actions (Marino et al., 2018). Other works, like Hafner et al., 2019 restrict the bit-depth of the images, yielding log-likelihood results that are not directly comparable with ours. Still other works, like SAVP (Lee et al., 2018), apply combinations of lower bound and adversarial losses. To our knowledge, the most similar recent work to evaluate log-likelihood performance is VideoFlow (Kumar et al., 2019). While their quoted performance is significantly higher than our models, their models are also substantially larger, consisting of 3 levels of latent variables, with 24 steps of flows between each level. In contrast, we use 1 or 2 levels of latent variables, we only 1 or 2 steps of flow. We have noted this in the updated draft.

We focused on log-likelihood as a metric of model performance, choosing video data for its ability to visualize aspects of autoregressive flows. Importantly, quantitative and qualitative metrics of images are not always well-aligned (Theis et al., 2015). Indeed, Kumar et al., 2019 note that there is only a weak correlation. While we agree that employing the aforementioned techniques to improve qualitative metrics is a useful direction, we felt it was more important to establish performance improvements on a clear quantitative basis. We intend to run an even more comprehensive set of experiments before the camera-ready submission deadline to investigate these possible improvements.

—Main contribution and related work

We have updated the background section to include the references suggested by the reviewers, as well as to clarify the main contribution of the work relative to these previous works. In our updated draft, we clarify that “we demonstrate that autoregressive flows can serve as a useful, general-purpose technique for improving sequence modeling as components of sequential latent variable models. To the best of our knowledge, our work is the first to focus on the aspect of using flows to pre-process sequential data to improve downstream dynamics modeling.” While Kumar et al., 2019 apply flows to video data, they do so by applying flows separately within each time step. Their process attempts to remove spatial correlations to get to the base distribution. We, instead, apply flows across time steps, processing the current frame based on previous frames. This process attempts to remove temporal correlations.

—Quantitative evaluation of decorrelation

We have updated the draft with an additional quantitative analysis of the temporal correlation. Indeed, we find that temporal correlation decreases in all three cases, corroborating the qualitative results. We also present a plot in Appendix C showing this temporal correlation decreasing during training, suggesting that these flows gradually learn a simplified basis for downstream dynamics estimation. We would like to thank R2 for this useful suggestion.

—Generated samples

We have included sets of generated samples in Appendix C. While these samples do not remain sharp over long temporal horizons, they capture backgrounds reasonably well. As noted above, we did not employ the range of techniques used to improve sample generation in video modeling, instead focusing on quantitative metrics. We intend to run additional experiments to improve sample generation quality, such as adjusting sampling temperature (Kumar et al., 2019).

---

### Decision · Program_Chairs · 2019-12-19

**Decision:**

Reject

**Comment:**

The paper scores low on novelty. The experiments and model analysis are not very strong.